# Gamma Radiation Dose Measurement Using an Energy-Selective Method with the Help of a Drone

**DOI:** 10.3390/s22239062

**Published:** 2022-11-22

**Authors:** András Molnár

**Affiliations:** John von Neumann Faculty of Informatics, Obuda University, Becsi ut 96/b, H-1034 Budapest, Hungary; molnar@uni-obuda.hu or molnar.pub@uni-obuda.hu

**Keywords:** outdoor dose distribution measurement, radioactive radiation mapping with a drone, radioactive radiation measurement with drone

## Abstract

Several dose distribution maps were obtained using a gamma radiation detector mounted to a drone. Based on the results and experience of the experiments, the shortcomings of the system and the possibilities for further development were identified. The primary goal of the development was to create a more compact, easy-to-carry, and easy-to-install system with increased sensitivity, which was achieved by several different methods and their combinations. During the discrete measurement procedure, the aim was to decrease the detection threshold, +0.005 to +0.007 μS/h measured above the background radiation. The increase in sensitivity was based on the characteristic energy spectrum of radiative materials. We took advantage of the fact that the radiating samples do not evenly increase the amount of gamma radiation over the entire energy spectrum. During the processing of the measurement data, we performed a comparison with the background radiation in the vicinity of the energy peaks characteristic of the sample and its decay products. This provides a better signal-to-noise ratio, thus enabling a more sensitive detection procedure. An important feature of our method is that in the traditional intensity curve displayed as a function of flight time only noise is visible, therefore one cannot directly conclude the presence of the sample. However, our method is clearly able to identify the location of the searched source at a height of 8 m with a continuous flight speed of 2 m/s using a 500 μS/h activity sample (as measured at a distance of 0.1 m from the sample). The increase in sensitivity allows either a higher scanning height (approximately +1 to 2 m) or, in the case of the same aircraft at the same altitude, a larger area from one take-off. Of course, the scan height or scan speed can increase significantly if the activity of the source being sought is high. In our experiments, we used a natural uranium mineral (Autunite) with activity far below that of artificially produced isotopes. In the series of our experiments, we also covered the detection of several sources, which modeled the possibility of mapping scattered active sources. The main advantages of the system developed and presented by us over the survey procedures used in practice is that a large area can be mobilized easily, without the risk of a human operator in the field, and the survey of a large area can be carried out at a low cost. The purpose of the system is to detect the presence of the source and to localize it to such an extent that the localization can then be easily refined by manual or other ground procedures. As we do not aim for positioning accuracy by centimeter, standard GPS localization is sufficient for the measurements. During the measurements, the geographical coordinates are interpreted in the GWS’84 system. The coordinates of the latitude and longitude circles are also shown in this system in the figures presented.

## 1. Introduction

Gamma radiation is statistical in its nature. This means that the amount of radiation cannot be deduced from the detection of a single gamma photon. It follows that radiation is characterized by the number of events per unit of time. However, the need for fast and reliable dose rate determination is to be considered in the field. Its conventionally accepted method is to place instruments in the area, or measure with an instrument at a given geographical coordinate, depending on its sensitivity, but holding it at one point for a well-defined period of time. By analyzing the measurement results thus obtained, a radiation map of an area can be produced.

It is more convenient and in many cases safer to secure the radiation measuring instrument on a drone and use it to maneuver the drone to the measurement points. The problem, in that case, is the limited operating time of the drone, which does not make it possible to ensure the measurement time of traditional instruments, which can often take several minutes. The solution to the problem can be approached from multiple angles. Of course, the operating time of drones can be increased, but the weight of the detector can also be reduced, which indirectly also increases the operating time of the drone. Another way is to increase the sensitivity of the detector, which leads to a decrease in the time required for measurement. The third way is to change the measurement principle. Using scintillation detectors, it is possible not only to detect a gamma photon but also to determine its energy.

The spectrometers used to measure gamma radiation allow one to determine the energy of the detected gamma particle. By analyzing the data from the spectrometers, the nature of the radiation and the characteristics of the source material can be determined. A distinction can be made between the decay products of the source (Figure 1) [1]. In this way, the presence of isotopes accidentally created by human activities in the air can be detected with high confidence.

Utilizing an energy spectrum different from the background radiation, a selective detector can be made that has a higher sensitivity in the energy range characteristic of the studied isotope than in the entire spectrum. The higher sensitivity allows faster measurement, thus scanning a larger area when detected with a drone-mounted instrument.

The measurements were performed as shown in Figure 1 using an AtomNano 8 instrument. The AtomNano 8 instrument is based on a CsI (Tl) (Thallium Activated Cesium Iodide) scintillation detector, similarly to the individually manufactured detector. For visualization, the free downloadable software called Becqerel Monitor (BecqMoni2011) was used [1].

## 2. Fundamentals of the Process

In the case of sufficiently long sampling, the number of detections can be handled well statistically. Yet, due to the severely limited flight time, using the device mounted on the drone we can only measure for a short time in the vicinity of each terrain point. That is the reason why we were looking for a detector that gives a countable number of detections even within a shorter time frame. Our old detectors provided 1–2 detections per second, which is completely unsuitable for detecting the source. In the case of a large number of detections, the possibility of examining the spectrum characteristic of the sample is explained in diagram 1 and the accompanying explanations

We strived to show that the energy window analysis shows an increase in sensitivity even in the case of such small samples, which are created with a 1-s measurement.

Since, in the case of a given radionuclide, the number of events detected by the detector per unit time (in this case only the number of gamma photons detected per unit time) is proportional to the dose rate, we therefore implicitly equate the number of detected events to the concept of dose performance.

The energy peaks of several nuclides are visible in Figure 1. It can be clearly seen that comparing the experimental measurement with the background radiation of the experimental site does not generally increase the number of detected events as a result of the sample. At the characteristic energy levels of the nuclides in the sample, the number of detected events increases to a greater extent than in other parts of the energy spectrum in general. As a result, the peaks shown in Figure 1 are formed. If the evaluation of the data is not performed for the whole spectrum, but only in the vicinity of the radiation peak of the examined sample, the so-called signal-to-noise ratio is more favorable for detectability due to the larger increase in the number of detected events as compared to the background radiation.

Figure 2 illustrates a comparative measurement. The spectra were recorded with a scintillation detector. The duration of the measurement was 1 sec for both the background radiation and the tested sample. The graphs show the total number of events in less than 1 s. This value was 115 at the study site without a sample and 144 with a sample. If the detector is used as a particle counter only, the number of events increased by 29 due to the sample, which is a clear indication of increased gamma activity in the vicinity of the detector.

Analyzing the spectrum, it can be seen that using the sample, additional events were observed in the channels that were not seen in the background, that is two events without increasing to three with the sample. If the full spectrum of the detector signals is analyzed, the signal-to-noise ratio is as follows:(1)$$\frac{S}{N}=\frac{144}{115}=1.25\;$$

If we calculate signal-to-noise ratio data using the same measurement values as the increased amount of peak values due to the sample, then:(2)$$\frac{S}{N}=\frac{3}{2}=1.5$$

Although obviously a better signal-to-noise ratio is obtained from the peak data, due to the small number of measurements, it is practical to evaluate the data in a smaller environment of the peak. Based on Figure 1, the width of the apex of the peak can be determined by knowing the pattern sought.

If not the total number of events per unit time is recorded during the measurements, but only the number of events observed in the vicinity of the energy level characteristic of the sample, a more sensitive measurement method is obtained.

## 3. The Detector Applied

The project aims to map the gamma dose distribution of an area using a detector mounted on a light drone. We use well-known methods by adapting them to a drone. This fundamentally results in a low-mass detector with high sensitivity. Therefore, the goal is to find a sensitive detector in addition to its small mass. That is the reason why a CsJ(Tl)-based scintillation detector was chosen. We accept that there is a more sensitive crystal, such as NaI(Tl), but based on the data in the corresponding literature the scintillation characteristic of the crystal we chose better matches the sensitivity characteristic of the MPPC (multi-pixel photon counter) [2]. The detection principle is not new. Physically, however, the detector is a type especially manufactured for us for the needs of the project.

The measurement is based on a scintillation detector specially designed for this purpose. The planned size of the CsJ (Tl) (Thallium Activated Cesium Iodide) crystal was 20 × 50 × 50 mm, i.e., 50 cm^3^. After manufacturing, however, the actual size of the crystal became 21.5 × 50 × 50 mm. The scintillation events are detected by an MPPC module [3,4,5,6]. The MPPC has several advantages over traditionally used PMT (photomultiplier tube) sensors [7]. Given that minimizing detector weight in addition to small size was a key consideration during development. Another advantage of the device compared to PMT tubes is that it does not require high electrical voltage, and it does not need to be shielded against external magnetic disturbances [8]. As a disadvantage, it should be mentioned that like all semiconductors, the MPPC is also sensitive to temperature changes, which must be compensated electronically [9,10]. MPPC is an appropriate choice.

During the setup of the detector, the ambient background radiation was measured with a certificated instrument and was 0.082 μS/h where the detector indicated an average of 88 photons per second. From a practical point of view, this means that the developed scintillation detector can generate an energy spectrum of an average of 88 gamma particles per second by dividing the energy of the detected particles into 8190 discrete values.

The detector is integrated directly with a digital processing unit that saves the measured data to a portable memory card. 1 s is the shortest measurement time of the detector. During this time, a highly detailed spectrum cannot be produced, but the measurement method is sufficient for the procedure described in the chapter. At the same time, the spectral data can be integrated for a longer time and thus produce a more accurate spectrum. The compact detector also includes a GPS unit, so it provides the time stamp and geographical coordinates from the GPS together with the data measured and recorded on the SD memory card.

Figure 3 compares the sensitivity applied to the entire spectrum with the energy window sensitivity characteristic of the sample. The diagram intentionally illustrates the difference through a textbook case, since the well-known law does not require a deeper explanation, so it is sufficient to examine the difference between the two curves, which significantly demonstrates the favorable effect of energy selective measurement.

Static measurements were performed to evaluate the sensitivity of the detector. These were performed at distances of 1–20 m from the sample, at every meter. Each measurement lasted one minute, which meant 60, 1-s spectra. The particle number values shown in Figure 3 (y-axis CPS values) are the median of the aggregate CPS values for the data set 60 per row. The blue diagram in Figure 3 shows the CPS values over the entire spectrum as a function of distance from the sample. It can be seen that within 5 m of the detector data shows a CPS increase, but a significant deviation from the background radiation (dashed, red line) is only present within 4 m. This value is four times higher than those by detectors used so far [11]. A similar scintillation detector’s threshold distance was 1 m for the same sample. Figure 3b illustrates the sensitivity of the old GM tube system as well as the old, small-sized (13 × 13 × 47 mm CsI (Tl)) scintillation detector. It is clear that meaningful measurements can be made with the sample used in the test at a distance of less than one meter. Figure 3a shows the sensitivity of the new scintillation detector measured under the same measurement conditions. It can be seen that even in particle number counter mode (blue diagram) the substantive measurements have been increased to a distance of 3 m. By taking advantage of the scintillation detector’s energy resolution with the selective energy window process, that distance increased to 5–7 m.

Given that the experiment does not aim for quantitative detection but only searches for the relative activity deviations of the test location, it is not necessary to use a calibrated device. During the experiment, the stability of the device is important, but its calibration is not relevant. In its entirety, the article processes the channel data of the device, and the device processes channel data. Of course, the channel values logically correspond to the gamma radiation energy. A function “f” creates the connection between the channel number and the energy, the coefficients of which can be determined with sufficient calibration, but from the point of view of this experiment, this is not necessary precisely because of the relative comparative analysis.

Given that the new detector allows not only to register the number of events but also provides energy resolution, we also examined the effect of using the selective range. Figure 4 displays the channel spectrum of the background radiation (orange diagram in Figure 4) as well as the spectrum of the sample (blue diagram in Figure 4). One can see that the Autunite sample results in an uneven increase in the number of events over the entire spectrum. Sample-specific peaks appear in the spectrum, two of which are marked in Figure 4.

The orange diagram in Figure 3a is based on the event numbers in the channel spectrum given in Figure 4. For better comparability, the event numbers measured in the two-channel ranges indicated were scaled to the event numbers for the entire spectrum. It can be seen that according to the orange curve, the effect of the sample can be detected within 8 m. Within 7 m, the increase is already considered significant.

The static measurement supported the theoretical approach formulated at the beginning of the paper, according to which the sensitivity of the detector can be improved by selective processing of the detected radiation.

Using the blue diagram shown in Figure 4, we determined the channel window in which the sensitivity of the measurement to the tested sample is higher. Accordingly, the 800–970 channel range used during the later practical field measurement was selected.

## 4. Measurement Procedure

The flight speed has a direct impact on the time spent in the vicinity of a given terrain point, which ultimately determines the number of detected events for a given detector. Respectively, the slower the device flies, the more interpretable the data that can be collected. However, due to the severely limited flight time of the aircraft, the reduction in flight speed significantly reduces the size of the area that can be covered with one take-off.

There are two reasons for the flight speed of 2 m/s. The first is that the DJI’s control system does not allow a lower speed. Naturally, we can bypass this using our own software, as we did in a previous series of experiments [5]. We tried to take the flight speed of 2 m/s as a compromise value.

The measurement procedure can be split into two phases. i.e., data collection and data processing. During data collection, a multicopter equipped with a detector performs a systematic scan flight over the study area. Since the flight takes place autonomously, relatively low, four reliable control systems ensure precise height and position maintenance [12]. The flight speed is 2 m/s. In contrast to the accurate method of discrete measurement [5], the scan flight is continuous. During the flight, the detector records the gamma energy spectrum every second as well as the GPS coordinates at the time of recording. More detailed GPS data will be stored in a separate file for later data analysis. The GPS used has a 5 Hz data update, so the GPS log file records 5 coordinates data per second and stores data describing the positioning conditions, such as altitude, satellite data used for measurement, localization accuracy, ground speed, satellite data, and timestamp from satellites. No special solutions are necessary for accurate location determination, as the final localization of the source is carried out by ground units (vehicle, robotized all-terrain vehicle, or human) based on the data provided by the drone [13]. The individual data files’ (GPS.log, gamma spectrum, detector.log) can be linked using the timestamp from satellites, which, later on, allows one to analyze the data from different aspects. The scanning flight is performed in automatic mode according to a pre-planned program, which ensures a constant altitude and uniform speed during the measurement. If the test area is larger than the multicopter can fly in one take-off, the measurement can be performed with more take-offs. In such cases, the data sets recorded during service time (landing, battery replacement, take-off) must be deleted during processing.

In the second stage of the measurement, the data recorded during the flights are processed offline, using software developed by us, and the result is visualized. The special software provides a complex analysis of the data of the new detector. The software can be used to perform basic data cleaning. This is important during the start of the measurement when, due to possible transient phenomena, or in the case of recordings from multiple take-offs, an outlier data set is created, or sections appear where the measurement path is left or includes take-offs and landings. In such cases, the software can delete unnecessary or erroneous data sets.

In order to be as flexible as possible, it is possible to process the data over the entire spectrum or within a user-designed energy window. An additional feature of the software is the various filters and highlighting of the data. With the preprocessed data, the software creates a dose distribution map with the help of which the activity of the examined area can be analyzed in a visual form.

In the first step of generating the dose distribution map, the measurement area is divided into subareas of the same size as shown in Figure 5:

Consider $\Omega_{i,j}\subset R^{2}\;X_{min}\;to\;X_{max}\;\mathrm{and}\;Y_{min}\;to\;Y_{max}$ an enclosed rectangular area, where
(3)$$X_{min}=\underset{p\in P}{\min}p_{x}$$
(4)$$X_{max}=\underset{p\in P}{\max}p_{x}$$
and
(5)$$Y_{min}=\underset{p\in P}{\min}p_{y}$$
(6)$$Y_{max}=\underset{p\in P}{\max}p_{y}$$
where p is the GPS coordinate of a measurement point ($p_{x},p_{y}\in R$), and P is the set containing all points. The n×m is formed as follows:(7)$$\Omega_{ij}=\{\begin{array}{l} \left[x^{i},x^{i+1})\times [y^{j},y^{j+1})\;if\;i\;\in \;[0,n,\ldots ,n-2]\;and\;j\;\in \;[0,m,\ldots ,m-2\right]\\ \left[x^{i},x^{i+1})\times [y^{j},y^{j+1})\;if\;i\;=\;n-1\;and\;j\;\in \;[0,m,\ldots ,m-2\right]\\ {}[x^{i},x^{i+1})\times [y^{j},y^{j+1})\;if\;i\;\in \;[0,n,\ldots ,n-2]\;and\;j\;=\;m-1\\ \left[x^{i},x^{i+1})\times [y^{j},y^{j+1})\;if\;i\;=\;n-1]\;and\;j\;=\;m-1\right] \end{array}$$
where
(8)$$x^{i}=x_{min}+i\times dx$$
and
(9)$$y^{i}=y_{min}+i\times dy$$

Based on the above, we determine a function f(p) that gives the value of point p. The value for the i,j-th cell can be determined using the following relation:(10)$$\bar{p}\Omega_{i,j}=\frac{\begin{array}{c} \sum^{\;}\;\;\;\;f\left(p\right)\\ \begin{array}{cc} {}^{p\in \Omega_{i,j}} & \end{array} \end{array}}{\left|\left\{p:p\in \Omega_{i,j}\right\}\right|}$$

The averaging function can be replaced by any other function, e.g., median. That is, the measurement area is divided according to the predefined parameters. All measurement points belong to a given cell. The value of a cell is given by the average, median, or maximum of the measured gamma radiation values, depending on the setting.

The last step of the display is to color the area units (cells) defined above with the values calculated for them according to the palette specified by the user.

## 5. Processing Software

A special software has been developed for fast and efficient data processing. The software basically implements three main interfaces that fit the three basic data processing procedures. Another purpose of the software is to visualize the results. In many cases, it is useful and can be interpreted more easily by the user if the processed data can be viewed not only numerically, but also in image form [14,15,16].

In the first step, the software is able to read the data from the special detector without conversion and place the measurement points on a map based on the data. For convenient operation, the map data are loaded by the software through a Google app, so that the location of the scanning flight and the path generated by the data recorded during the scan can be checked on a real satellite image (Figure 6). The interface facilitates deleting unnecessary and potentially confusing flight sections. These include pre-take-off preparations when the system is ready and recording data, but the device is not flying. Near-ground measurements usually result in higher detection values and should not be considered in conjunction with scanning data. Measurements from multiple takeoffs can be similarly confusing when data are recorded during battery replacement.

In the second step, the data are represented in a Cartesian coordinate system (Figure 7). The “x” axis contains the channel number of detected events (the energy level after calibration), while the “y” axis contains the serial number of the measurements. Accordingly, 8190 values belong to one row, which corresponds to the distribution of events detected within a given second according to the energy level (number of channels). The graph also displays a vertical graph as a function of time, which assigns all detection numbers to the given measurement second. In fact, this diagram corresponds to the classic time/event number or, in other words, the time/gamma dose diagram. As the experiments are aimed at detecting low-dose samples, the chart does not show any protrusion, i.e., an increase in activity. At the same time, the diagram is suitable to visualize the disturbances recorded during the measurement (for example, a power-on transient or other possible data storage error) and to exclude them from further processing. On the interface, it is possible to specify the data ranges that we want to take into account when calculating the dose distribution for further processing. The two vertical lines in Figure 7, which mark the lower and upper limits of the processing window, serve this purpose. By default, this applies to the entire channel range (energy range), but we can also set a narrower energy range for the sample you are looking for. There are two ways to modify native data. The first option is filtering, during which the effect caused by noise can be reduced. To filter the data, the software ensures averaging in a specific sliding window (moving average) or median filtering in the same sliding window. The second option is practically just a highlight feature to enhance the visual experience. Highlighting means increasing or decreasing the data using a function. Modifier functions can be parameterized as power or exponential functions.

In the third step, a dose distribution map of the measurement area is created (Figure 8). The map is generated based on the measurement data set and possibly modified in the previous step. In this section. the resolution of the map can be set. The resolution is interpreted in Figure 7, i.e., how many coherent units the map can be divided into. This value typically means squares between 1 and 4 m of side length. The side lengths of the squares depend on the parameters of the scan path. It’s a good idea to specify cell sizes during rendering that fit well with the actual scan. For example, if the parallel tracks of the scan flight were 3 m apart, it is practical to split the display into squares with a side length of 3 m. The “x” axis of the map corresponds to the geographic longitude circle, while the “y” axis corresponds to the latitude circle. The coordinates are interpreted in the WGS 84 projection system. To more accurately identify any possibly clean-cut source or other areas to be identified, the interface provides the user with a movable crosshair. Placing this crosshair on the examined point makes its coordinate readable.

Figure 8 shows the location of a strong source indicated by a yellow square. On the right side of the figure, one can read the difference, or in other words, the contrast between the source and the environment on a color scale. The recording took place during a low-altitude flight at an altitude of 2 m above the source. The 2-m distance provides strong and reliable signals for the detector, so a good reference measurement can be achieved.

## 6. Practical Experience

In order to test the method in practice, the detector was mounted on a DJI Inspire I quadcopter [17]. The quadrocopter is able to fly autonomously on a pre-programmed route. Measurements were performed in a test area 27 m long and 24 m wide. The scanning flight took place at an altitude of 8 m above the ground, at a steady horizontal speed of 2 m/s. The radiation source was the autunite mineral used in previous experiments [11,18,19]. Autunite is a natural mineral of uranium. With its application to the measurement process of natural geological enrichments, it provides many empirical results [20,21].

The measurement data were processed directly by the presented software. The filtering and highlighting parameterization was set based on the data of the measurement results obtained during the static detector test during processing. Accordingly, the channel window has been set to the channel range 800–970 according to Figure 4. In this range, data were highlighted by a factor of 20.

Figure 9 illustrates the two kinds of processing results of one measurement. The figure on the left is based on processing the data without filtering as if the detector had recorded only event numbers. Thus, according to the particle counting mode, the number of particles detected in the full spectrum was calculated. It can be seen that no source can be clearly identified in the image on the left in Figure 9 (one source was placed in the test area during the measurement).

The right-hand image of Figure 9 illustrates the energy window determined in the previous measurements (in this case the channel shape), highlighting the values therein with a linear multiplier. It can be seen that in this case, one distinct area on the map is clearly seen that is close to the actual location of the outsourced source.

The location of the reference source is illustrated in Figure 8. The essential difference between the reference source and the area scan analysis is that the reference source was measured at the end of a post-scan flight during a static suspension. So, the drone did not move while recording the source data. During the experimental measurement, however, the drone traveled at a constant speed of 2 m/s during the entire measurement. Since the data were recorded every second, a coordinate shift of 1–2 m is generated during the measurements. The source position error in Figure 9 is 3 m. Since the spatial distribution of the processing was defined in 2.5 m cells (a colored cube on the map 2.5 m × 2.5 m), the position error is within the error limit due to motion and processing.

## 7. Conclusions

The energy-selective processing procedure improved the detectability of the outsourced sample as expected. As presented in the dissertation, the sample was also detected in a measurement environment when its signals were lost to the noise caused by the background radiation. The software developed for processing allowed for convenient and fast analysis. The detector developed for the task worked flawlessly. Its size (weight and volume) is suitable for use onboard various drones. The 400 g weight of the detector is especially remarkable. Many other applications [22,23,24,25] use a detector of nearly 4 kg with similar sensitivity, which is a significant flight time constraint for small drones. The detector is very easy to operate, no special expertise is required.

Based on experiments using similar systems [26,27], our measurement method is simpler and does not require several measurements at different heights above the area. The resolution of the detector we used is nearly eight times of the BGO crystal detector in the referenced experiment.

In the future, it is necessary to develop protocols that can be used to quickly set up the processing software. It is necessary to determine the lower and upper energy values of the ideal energy window in the light of the energy spectrum of the radiant material sought.

Further measurements are required with other isotopes and sources that mimic a real contaminated area [28,29].

Our experiments have confirmed the findings in other studies [30,31] that monitoring low-altitude radioactivity with drones is a beneficial and cost-effective method.

## Figures and Tables

**Figure 1 sensors-22-09062-f001:**
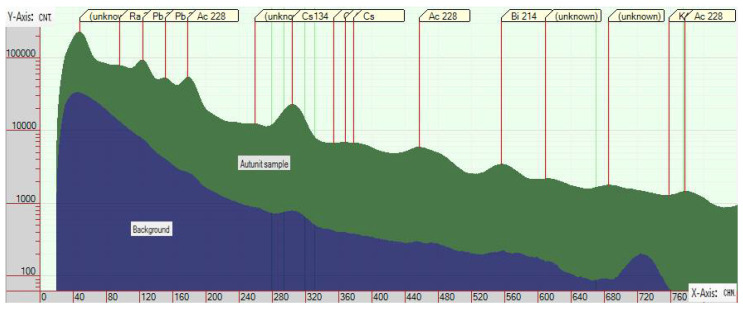
Comparison of the radiation spectrum measured at the test site with that of uranium ore.

**Figure 2 sensors-22-09062-f002:**
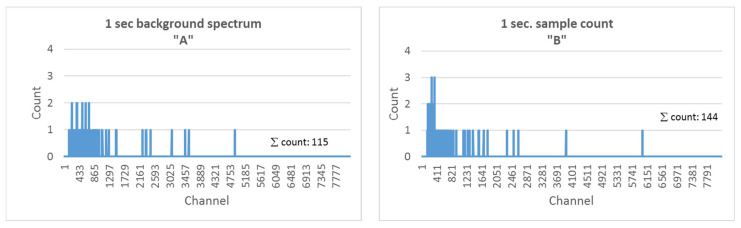
Background radiation (**A**) and the low-activity sample (**B**) of a 1-s measurement.

**Figure 3 sensors-22-09062-f003:**
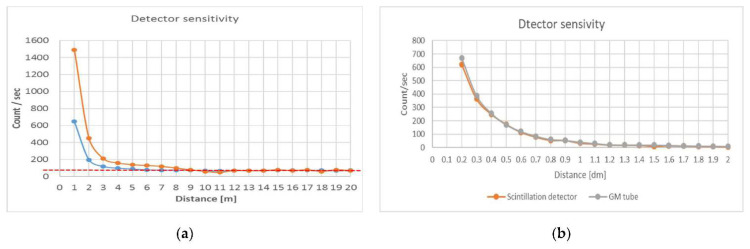
Static test of new detector (**a**), and old (**b**) sensitivity. The orange line whit selective energy window, blue line without selective energy window.

**Figure 4 sensors-22-09062-f004:**
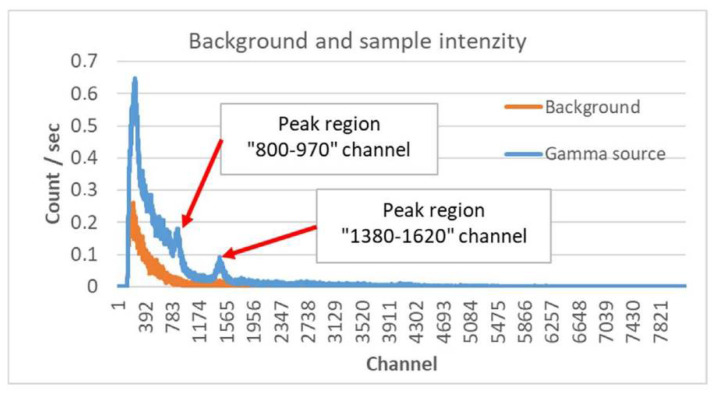
The spectrum of the static sensitivity test of the detector for background radiation (orange) and Autunite source (blue).

**Figure 5 sensors-22-09062-f005:**
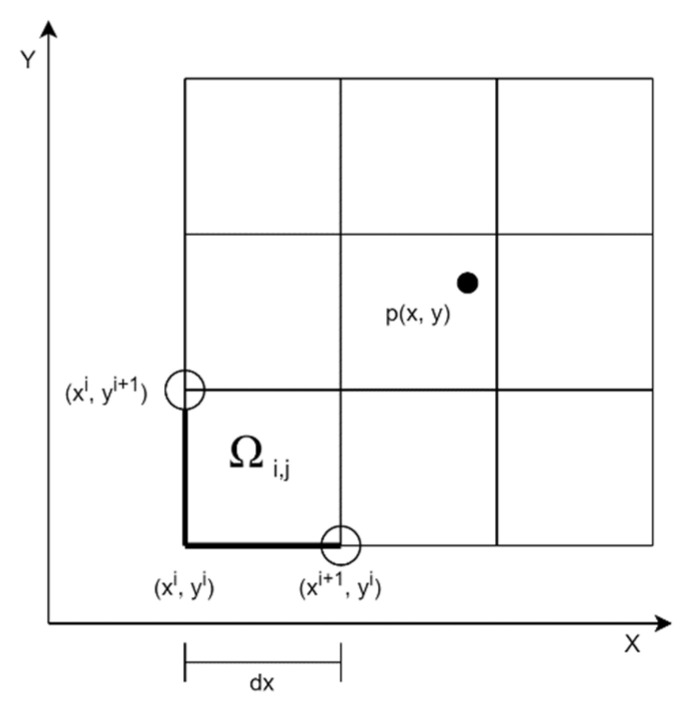
Dividing the area into cells.

**Figure 6 sensors-22-09062-f006:**
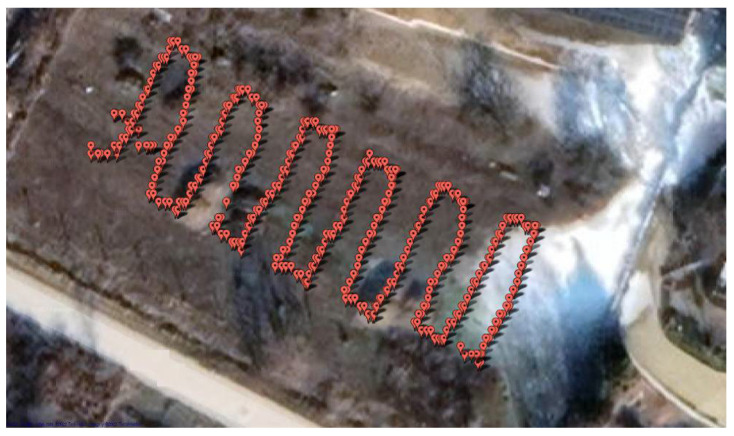
The screen of the processing software displaying raw measurement points.

**Figure 7 sensors-22-09062-f007:**
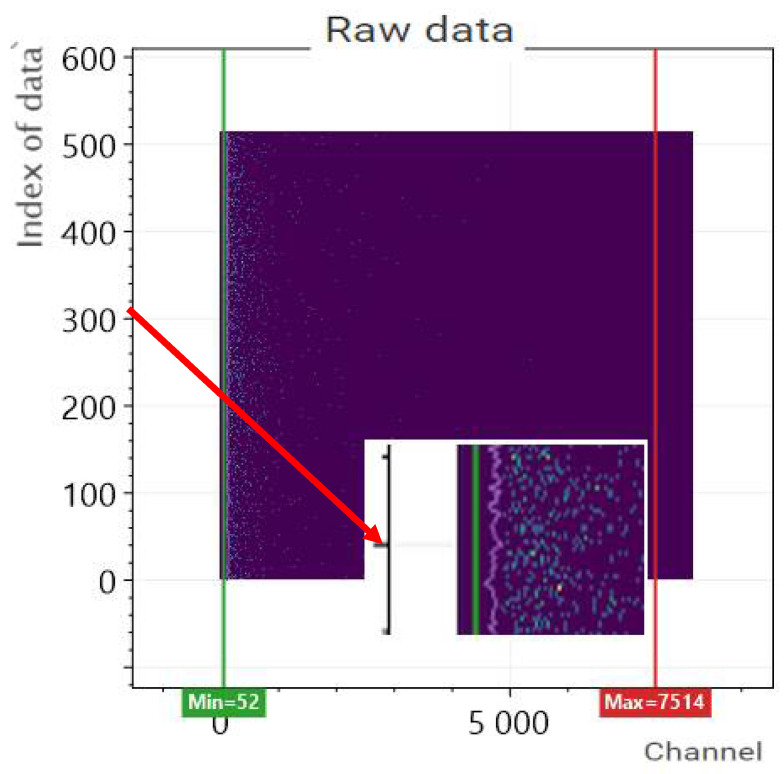
The monitor screen of the measured data obtained by the processing software.

**Figure 8 sensors-22-09062-f008:**
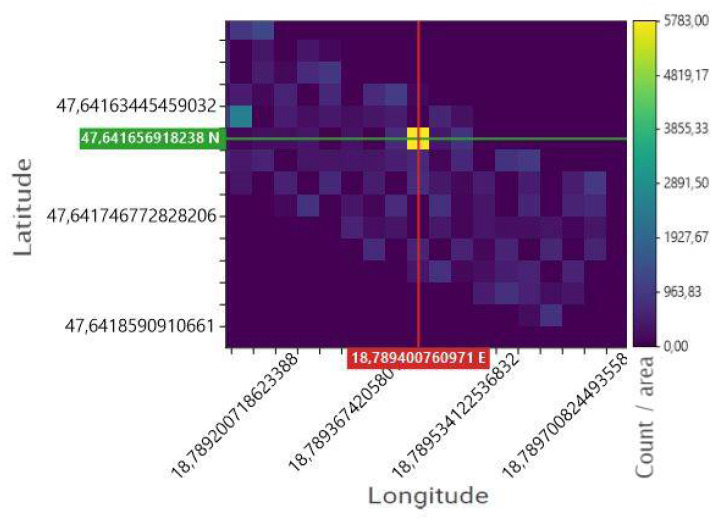
Dose distribution map generated by the processing program.

**Figure 9 sensors-22-09062-f009:**
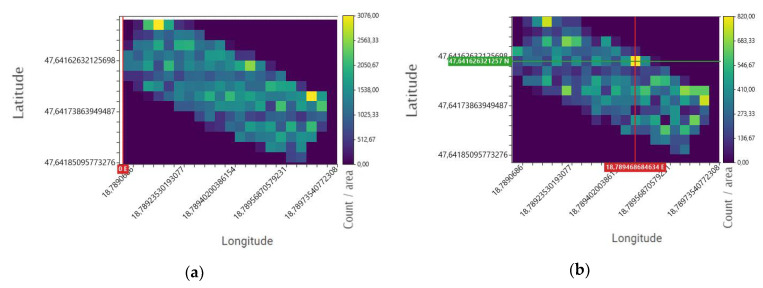
Shows the dose distribution map generated by the processing program without an energy window ((**a**) panel) and using an energy window ((**b**) panel).

## Data Availability

Not applicable.

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
