# Peer review of "Gamma Radiation Dose Measurement Using an Energy-Selective Method with the Help of a Drone"

_sensors, 2022, doi:10.3390/s22239062_

Round 1
Reviewer 1 Report
I have thoroughly read the manuscript titled “Gamma radiation dose measurement using an energy-selective method with the help of a drone” by András Molnár. The work is important and worth investigating. The manuscript is well written. The introduction provides sufficient background but includes vague references. It is highly recommended to add more relevant references for the interested reader to have an understanding of the field.
The research design is appropriate, and the methods are adequately described. Furthermore, the results are clearly presented, and the results support the conclusion.
It is recommended to remove unnecessary details from the abstract section and only include the main results here.
After carefully considering the previous comments, the manuscript may be considered for publishing in the journal.
Author Response
Dear Sir/Madam,
I corrected every mentioned case in my article.
Reviewer 2 Report
The possibility to use a drone mounted sensor to obtain a gamma radiation map is studied in this work. It is an interesting proposal, with a clear usefulness potential, although from my point of view the experimental design is not clearly explained enough. Also, I think that it would be a good idea to include a deeper literature review on this topic, and compare in more detail the system and results obtained in this work to those found in the literature, and also to other similar works from the same author.
After a relevant rewriting of this manuscript, I think that it might be interesting to publish this work, but several points should be improved first:
1. There exist a lot of similar proposals that have been recently published (just check "gamma" + "maps" + "drone" in any specialized index and a lot of works appear), but almost none of them are cited in this work. The bibliography is too short and not discussed in detail. Refs [3] and [8] are actually the same, and 3 of the 7 references are from the same authors from this work.
2. Related to the previous point, I would expect a much deeper comparison to the same author previous works. In which way that this current work improve or provide new results in comparison to the previous works? Reference [5], published in this very same journal is actually really interesting, and from my point of view, more detailed and complete that this manuscript. I would take that paper as an example on how this manuscript should be, and clearly state how both of the differ and in which way this manuscript includes new results in comparison to that one.
3. In several parts of the manuscript, starting in the abstract, the author states that this proposal provides "better signal-to-noise ratio", or an "increase" in sensitivity. Such statements make sense if two alternative systems are compared, but it is not clear anywhere to which system is the author comparing the current results. Is it maybe a comparison to the proposals for his own previous works? In any case, if it is said that this system improves the obtained results, it should be crystal clear to which alternative proposal is compared to.
4. In the abstract, in lines 12-13, it is said "the aim was to decrease the detection threshold +0.005 to +0.007 $\micro S / h$". I image that there must be a typo here, given that moving from 0.005 to 0.007 would be an increase of the threshold, not a decrease. In any case, this is the only reference to such threshold in the entire work so it is impossible to know what the author means and for this reason this point should be discussed in detail within the text.
5. One of the key points of this work seems to be the proposal to use a specific window of the spectrum for a specific radionuclide, in comparison to using the entire spectrum. It is my understanding that this proposal is justified in section 2, but from my point of view the data used in this section is not relevant enough to support such choice. Given that the measurements in this section last only 1 second, the number of detected events is too low. The uncertainty of the values in equations (1) and (2) should be considered. The relative uncertainty in equation (1) is around 20%, and in equation (2) is around 125%, making the differences between two numbers not significant from a statistical point of view. And this is only considering the counting uncertainty $\sqrt(N)$ with k=1.
6.The effect of the flying altitude is discussed with some detail, and arguments are provided to support the final height selection. I think that a similar approach should be followed for the flying speed. It is indeed said that it is possible to choose a different speed but it is not clear to me why a speed of 1 m/s is better than other. For lower speeds more counts could be detected and the uncertainty would be lower.
7. I think that there is some relevant technical information missing in the manuscript. Which MPPC was used? Which are their characteristics? Which certified instrument (lines 99) was used to measure the background dose rate?
8. I don't understand how the scaling mentioned in lines 132-133 is performed.
9. Throughout the paper it is said the gamma dose rates are measured, but from my understanding only total number of counts or count rates are measured.
10. From my point of view more details should be provided about the filtering and highlighting functions of the software, mentioned in liens 226-230.
11. In figures 8 and 9 the axis are labelled with coordinates using an unnecessary large number of significant digits. Also, the numbers that appear in the color scale could be chosen to be round numbers, there is no need to use so many significant figures.
In summary, although the system proposed in this work seems interesting, I think that the manuscript should be further improved to clearly explain their novelty compared to other works, and to provide more details on the experimental setup and improve the results presentation and discussion. For these reasons I cannot recommend its publication in Sensors in its present form.
Author Response
Dear Sir/Madam,
I attached my detailed answers.

Reviewer 3 Report
My impression of the article is as follows:
The author took an already known problem and a set of elements, replaced the elements with other ones with obviously low sensitivity (compared to the works in the references, e.g. NaI(Tl) in [3], has much higher sensitivity).
The authors did not specify the technical characteristics of their measuring device, claiming that they increased the sensitivity. I did not find evidence of this in the article. The author states about the advantages of the selected devices without references or with inappropriate references.
Problems of this kind cannot be solved by experimental means only, it is necessary to simulate the radiation transport for different characteristics of the geological environment. The authors have considered only one possible background source (spectrum).
In fact, the soil (geological environment) can be highly heterogeneous and if the content of natural radionuclides, which the author mentions as "background values", in a certain part of the studied area will be higher than in others, then such data can be interpreted in the described by the authors site as a radioactive source.
Figure 3 is a standard student paper on dosimetry, I recommend deleting it, as it is a well-known law.
Paragraphs «The static measurement supported the theoretical approach formulated at the beginning of the dissertation, according to which the sensitivity of the detector can be improved by selective processing of the detected radiation» and «As presented in the dissertation, the sample was also detected in a measurement environment when its signals were lost to the noise caused by the background radiation.» (Line 137 and 290) indicate that the authors treated their article with disrespect, to put it mildly, ripped the above phrase out of the dissertation and did not even bother to rephrase it.
Figure 4 is incomprehensibly described, the natural radionuclides in the background radiation and their activity are not indicated, as well as what are the 2 peaks in the source and their energy is not indicated. In this regard, Figure 4 cannot be evaluated by readers.
It is important to note that in all figures of the spectra the channel number is used, but the energy is missing, this indicates to me that the energy calibration of the spectrometer was not carried out. This makes it very difficult to assess the adequacy and significance of the results.
Section 4 in this one is a repetition with a slight paraphrase from the author's previous article (reference [5], section 3). I recommend that it be deleted entirely, making reference to the authors' previous article.
Especially in Figure 9a, but also in b) you can find 2 specimens on the edges of the study site. Hopefully these are not the 2 samples from the previous article [5]. However the author declares that the method works perfectly (Line 293) without justifying the choice of the channel range 800-970 (energy should be presented instead of channel in the figure; a spectrum of the investigated sample is missing).
English needs to be corrected in some places in the text, check the excessive use of "sought".
The abstract needs to be shortened. The introduction makes absolutely no sense and does not present a literature review, so it needs to be redone. The origin of Figure 1 is also unclear (reference [1] is a program).
Chapter 2 is also nothing new and I think it should be deleted.
Chapter 3 should be called "Materials and Methods" and partially include the contents of chapters 2,5 and reference to chapter 4, since it is borrowed.
Chapters 5,6 are "Results and discussion," which should be expanded nevertheless.
Chapter 7 should be called "Conclusion".
The list of references should be expanded and all references should be on the topic.
Bottom line:
Major revision, because a significant redesign of the article is needed: improving the description, restructuring, etc.
Other comments and remarks are in the attached PDF file.

Author Response
Answers attached.

Round 2
Reviewer 2 Report
Although I recognise that the authors has made some efforts to improve the manuscript and to answer my comments, I think that most of them still remain withouth many changes. The author did not always specify in detail in his response where the changes where included, so it is possible that I did not detect all of them.
To give some more detail I mention the following points:
- Although the bibliography does indeed include more references, the introduction is still too short and does not provide a propper introduction to this field of work. (point 1 in my previous report)
- I do not think that the comparison to the previous authors work has been improved (point 2).
- The claims about an increased performance and sentitivity of this proposal, should be accompanied by the evidences to support it next to the claim, not in a single sentence hidden within the paper (point 3).
- In the decrease in the detection threshold is actually $0.005 \pm 0.007$, then it is not significant and all conclusions obtained from this value have no evidence to support it. (point 4)
- I'm not conviced with the answer provided to my previous point 5. I still think that there is lack of evidence to support the claims made by the author.
- I do not find in the new manuscript version the information that I asked in point 7 about the MPPC details and the instrument used to measure de backgrund dose rate.
- Point 8 remains unanswered.
- The problems with the axis labels in the figures should not be solved "covering number", but using a propper ploting solftware or learning how to properly use it.
In summary, I do not think that my previous questions and or suggestions have been properly handled and for this reason I still think that this manuscript should not be published.
Author Response
Dear Reviewer,
I am seeking to respond to your written observations and requests in the modified version of the material. I ask for understanding from the honorable reviewer on two points:
• The article did not aim to create a new detector or principle. The purpose of this article is to present a measurement procedure that enables gamma radiation measurement using a small drone, which is very much limited in lifting capacity and flight time. For this purpose, based on our experience so far, I requested the production of a detector that is significantly more sensitive to background radiation compared to the small, therefore light detectors we have used so far. This condition was basically met by a detector equipped with a sufficiently large scintillation crystal. The foreign manufacturer produced the device for me. Its internal design is not known to me and is not important. Due to the war situation, I cannot reach the manufacturer, so I cannot provide the type of sensor in question. I do not wish to disassemble the detector itself.
• I do not know if the reviewers can see each other's reviews. Unfortunately, I cannot fulfill conflicting requests. In such cases, I seek to strike a balance, for instance between expanding the introduction (one expectation), deleting the introduction (another expectation) and it being just right (third expectation).
I tried to fulfill the request for comparison. In the article, I included a diagram illustrating the sensitivity of the old detectors and pointed out the improvement in the sensitivity of the new detector.
I added an additional short explanation to the article for the reason of defining the highlighted channel range of the measurement.
I trust that the new version of my uploaded article is acceptable.
Reviewer 3 Report
You have not considered improving article.
Mine and other reviewers comments were ignored and not addressed in current revision.
If you can not see my comments (there are a lot of them) in PDF file, please consult IT specialist (or open PDF in your browser, preferably Firefox).
Because there were made not significant changes, my comments and suggestions still are valid.
You can find my comments (PDF) at previous review report.
Author Response
Dear reviewer,
I am seeking to respond to your written observations and requests in the modified version of the material. I ask for understanding from the honorable reviewer on two points:
• The article did not aim to create a new detector or principle. The purpose of this article is to present a measurement procedure that enables gamma radiation measurement using a small drone, which is very much limited in lifting capacity and flight time. For this purpose, based on our experience so far, I requested the production of a detector that is significantly more sensitive to background radiation compared to the small, therefore light detectors we have used so far. This condition was basically met by a detector equipped with a sufficiently large scintillation crystal. The foreign manufacturer produced the device for me. Its internal design is not known to me and is not important. Due to the war situation, I cannot reach the manufacturer, so I cannot provide the type of sensor in question. I do not wish to disassemble the detector itself.
• I do not know if the reviewers can see each other's reviews. Unfortunately, I cannot fulfill conflicting requests. In such cases, I seek to strike a balance, for instance between expanding the introduction (one expectation), deleting the introduction (another expectation) and it being just right (third expectation).
I tried to fulfill the request for comparison. In the article, I included a diagram illustrating the sensitivity of the old detectors and pointed out the improvement in the sensitivity of the new detector.
I added an additional short explanation to the article for the reason of defining the highlighted channel range of the measurement.
I trust that the new version of my uploaded article is acceptable.
Round 3
Reviewer 2 Report
I find the current version of this manuscript suitable for publication. I only have one small comment for the author, related to a typo:
1. In the new figure 3b, the title is "Dtector sensitivity" instead of "Detector sensitivity"
I do recommend then the publication of this manuscript in its present form in the MDPI Journal "Sensors".